# Benchmark Study on Phosphorescence Energies of Anthraquinone Compounds: Comparison between TDDFT and UDFT

**DOI:** 10.3390/molecules28073257

**Published:** 2023-04-05

**Authors:** Yujie Guo, Lingyu Zhang, Zexing Qu

**Affiliations:** Institute of Theoretical Chemistry, College of Chemistry, Jilin University, Changchun 130021, China

**Keywords:** phosphorescence energies, time-dependent density functional theory (TDDFT), anthraquinone

## Abstract

Phosphorescent material is widely used in light-emitting devices and in the monitoring of cell phenomena. Anthraquinone compounds (AQs), as important phosphorescent materials, have potential applications as emitters for highly efficient organic light-emitting diodes (OLEDs). Therefore, the accurate calculation of the phosphorescence energy of anthraquinone compounds is particularly important. This study mainly analyzes the phosphorescence energy calculation method of anthraquinone compounds. The time-dependent density functional theory (TDDFT) and the unrestricted density functional theory (UDFT) with seven functionals are selected to calculate the phosphorescence of AQs, taking the high-precision coupled-cluster singles and doubles (CC2) method as a reference. The results showed that the mean unsigned error (MUE) of UDFT was 0.14 *eV*, which was much smaller than that of TDDFT at 0.29 *eV*. Therefore, UDFT was more suitable for calculating the phosphorescence energy of AQs. The results obtained by different functionals indicate that the minimum MUE obtained by M06-2X was 0.14 *eV*. More importantly, the diffuse function in the basis set played an important role in calculating the phosphorescence energy in the M06-HF functional. In the BDBT, FBDBT, and BrBDBT, when M06-HF selected the basis set containing a diffuse function, the differences with CC2 was 0.02 *eV*, which is much smaller than the one obtained without a diffuse function at 0.80 *eV*. These findings might be of great significance for the future study of the phosphorescence energy of organic molecules.

## 1. Introduction

Purely organic phosphorescence emitting molecules have attracted tremendous attention in the past few decades due to their advantages of great variety, low price, and ease of fabrication [1,2,3,4]. In general, phosphorescent materials need an efficient intersystem crossing (ISC) rate, and the metal-free molecules, which usually have a small spin–orbit coupling (SOC), struggle to emit phosphorescence [5,6,7,8,9,10]. However, EI-Sayed’s rule states that the ISC rate could be relatively large if the radiationless transition involves a change in molecular orbital type (for example, from a (π, π*) singlet state to a (n, π*) triplet state and vice versa [11]). This rule is very useful in designing purely organic phosphorescence molecules. Recently, it has been suggested that anthraquinone compounds (AQs) have a fast ISC rate and can be used as the important components for metal-free phosphorescent materials [12,13]. The diversity of the electronic structure of Aqs has drawn more interest regarding research and development space in the fields of photochemistry. As a result, the detailed information on the low-lying excited states, as well as the phosphorescence emission of AQs, has become a popular issue in theoretical research [14,15].

The phosphorescence emission usually refers to the radiation transition from the lowest triplet excited state (T_1_) to the singlet ground state (S_0_), and the calculation of the triplet excited state is of particular importance in the simulation of the phosphorescence spectrum. The triplet excited states usually show the opposite electronic behavior when compared to the ground state, and the dynamic correlation should be included in the calculation of the triplet excited states. Thus, the coupled-cluster method, as a high level ab initio method that includes dynamic correlation, is usually adopted for the calculation of phosphorescence energy. Thiel and colleagues [16,17] tested singlet–triplet energy gaps for small organic molecules (limited to 15 atoms) with coupled cluster methods (CC3 [18] and CCSDR [19]). Although further calculations with complete active space with the second-order perturbation theory (CASPT2) [20,21] and coupled-cluster singles and doubles CC2 [22] can be extended to 40 atoms [23,24,25,26,27,28,29], these methods cannot overcome the “Index Wall” of high-level wave function theory (WFT). Density functional theory (DFT) [30,31,32] can obtain the energy and density of molecules in a way that is easy to calculate and DFT provides a balance between accuracy and computational cost for ground state calculations. Time-dependent density functional theory (TDDFT) [33,34,35,36,37] is the most widely used tool for calculating not only excitation energies, but also excited state properties, such as dipole moments and the emission spectrum, which is conducted with a computational cost between semiempirical methods and wave function theory. Latouche and colleagues benchmarked the phosphorescence spectra of transition metal complexes containing platinum and iridium with the unrestricted density functional theory (UDFT), time-dependent density functional theory (TDDFT), and the Tamm–Dancoff approximation (TDA) [38,39,40,41,42,43,44,45] methods. The results showed that UDFT and TDA performed better than TDDFT [46]. Recently, Ehara and colleagues used the symmetry-adapted cluster-configuration interaction (SAC-CI) [47] method and TDDFT to benchmark the geometric structures and phosphorescence energy of heterocyclic compounds; they concluded that both methods could provide accurate results in calculating phosphorescence energies of purely organic molecules [48]. Each of these methods has its advantages and limitations.

In the UDFT calculations, the singlet ground state and triplet excited state were optimized separately and could be easily converted for both states. However, in this approach, the Kohn–Sham orbitals were optimized and restricted to a different state; further, the phosphorescence energy was obtained at different levels. More importantly, the spin contamination of the triplet state from the unrestricted treatment usually underestimates the phosphorescence energy. On the other hand, the TDDFT treatment was free of spin contamination, and the triplet excited state and the singlet ground state can be calculated at the same level. However, the shortcoming of TDDFT is that the Rydberg states, which are caused by the Rydberg excitations, could increase the difficulty of the convergence for the triplet states [49,50]. Thus, the UDFT treatment, which has no Rydberg excitation, could be easier to converge for the optimization of the triplet state [51,52].Since the accuracy for all kinds of DFT calculations is also dependent on the choice of the functional, this work also investigates the functional and basis set effects on the calculation of phosphorescence energies with DFT. At the same time, in order to better calculate the phosphorescence energy and to determine the characteristics of the electron excitation, the charge transfer characteristics and the stabilization energy were calculated using the natural bond orbital (NBO) [53,54,55].

## 2. Computational Details

A series of 12 typical experimentally synthesized anthraquinone compounds and AQs, including 1-hydroxyanthraquinone (HA); aminoanthraquinone (AAT); benzophenone (BP); 5H-thieno[3,2–b] thioxanthen-5-one (TX-BT); 13H-benzo[4,5] thieno[3,2–b]thioxanthen-13-one (TX-DBT); dibenzo[b,d]thiophen-2-yl (phenyl)methanone (BDBT); ben-zo [b,d]thiophen-2-yl (4-fluorophenyl) methanone (FBDBT); (4-chlorophenyl) (dibenzo[b,d]thiophen-2-yl)methanone (ClBDBT); (4-bromophenyl) (dibenzo[b,d] thiophen-2-yl)methanone (ClBDBT); (4-bromophenyl) (dibenzo[b,d]thiophen-2-yl) methanone (BrBDBT); xanth-9-one (XA); 10H-acridin-9-one (AR); and thioxanthone (AD) [56,57,58,59,60,61,62,63]—as is shown in Figure 1—were used to calculate the phosphorescence energies. Here, the HA, XA, AR, and AD molecules were constrained to a C_s_ symmetry and the other molecules were in a C_1_ symmetry. For the molecules with C_1_ symmetry, the natural transition orbital (NTO) and the frontier orbital analyses were used to determine the wave functions of the lowest triplet excited state. Seven typical functionals of PEB, PBE0 [64], B3LYP [65], CAM-B3LYP [66], ωB97XD [67], M06-2X [68], and M06-HF [69,70]—ranging from GGA to meta-GGA functionals and cc-pVTZ—were used for the DFT calculations. In this work, TDDFT and UDFT were used to optimize the geometry and to compute the energy for the lowest triplet excited state, and thus the combination of the above two methods can yield the following groups of calculation schemes. If the geometries were optimized by UDFT and the excitation energies were computed with TDDFT, based on UDFT optimized geometries, then we denote this scheme as TD//UDFT. On the other hand, if both the optimized geometries and excitation energies were computed with UDFT, we denote this scheme as U//UDFT. In addition, there are other two kinds of schemes that were characterized as TD//TDDFT and U//TDDFT, in which the geometries were optimized by TDDFT and the excitation energies were computed with TDDFT and UDFT, respectively. The schemes are compared in Appendix A, and the results for TD//TDDFT and U//TDDFT were similar. All the geometry optimizations and single-point calculations with DFT were computed using Gaussian16 software [71]. The charge transfer characteristics and the stabilization energies were obtained using NBO Version 3.1 in Gaussian16 software. Besides DFT, the coupled-cluster method CC2 was used as the benchmark for calculation of the phosphorescence energies. In addition, all the CC2 results were obtained from the MOLPRO2021 package [72,73].

## 3. Results

### 3.1. Geometry Optimization

In this work, we calculated the phosphorescence energy; furthermore, the triplet state could be optimized by means of TDDFT and UDFT. On this basis, the root mean square deviation (RMSD) of the two geometries was calculated to measure the geometric deviation between them. Here, we aligned two geometries by visual molecular dynamics (VMD) [74], which was equivalent to minimizing the RMSD between the two geometries by translating and rotating one geometry.
$$\mathrm{RMSD}=\sqrt{\frac{1}{N}\sum_{i}\left[{\left(x_{i}-x_{i}^{\prime }\right)}^{2}+{\left(y_{i}-y_{i}^{\prime }\right)}^{2}+{\left(z_{i}-z_{i}^{\prime }\right)}^{2}\right]}$$

As shown in Table 1, based on the RMSD values, the average value of the 12 systems is 0.03 *Å*. This indicates that the geometry optimized by TDDFT and UDFT is very similar. Here, it should be noted that TDDFT suffers from a Rydberg state for HA, XA, AR, and AD, which caused problems in the optimization and electronic state selection, which was not easy to converge. In the optimization of TDDFT, we needed to choose the right state to track in order to obtain a reasonable geometry, not the lowest triplet state. UDFT is not affected by the Rydberg state during geometry optimization and was a more suitable method for optimization than TDDFT. The calculated phosphorescence energy values further show that the geometries obtained by TDDFT and UDFT optimization have little effect on phosphorescence energy calculation. As shown in Table 2 and Appendix A, the difference in the mean unsigned error (MUE) for phosphorescence energy was about 0.01 *eV*, further proving that the geometries obtained by TDDFT and UDFT had little influence on the calculation of the phosphorescence energy. Therefore, it could be concluded that the UDFT method was more suitable for the geometry optimization of phosphorescence energy.

### 3.2. Phosphorescence Energy

Table 2 and Table 3 show the triplet state geometry optimization by means of UDFT, and the phosphorescence energy was calculated by using TDDFT and UDFT, respectively. Compared with TDDFT, although the electronic states of UDFT were unreasonable, the lowest triplet excited state energy calculated by TDDFT was too low, making UDFT more suitable for calculating the phosphorescence energy. In most cases, UDFT is recommended to calculate the lowest triple excited state, whether it is a single point calculation or a geometric optimization. Although the spin contamination existed in the UDFT, it can be seen that the S^2^ values for the triplet states of AQs are around 2.0 for all of the functionals (as shown in Appendix A). Thus, the effect of spin contamination on the results is small.

Furthermore, it could be concluded that the phosphorescence energy obtained by TDDFT was underestimated and all the values were lower than those found in CC2. As the calculation of DFT was affected by the functional, the analysis of different functionals showed that PBE without the Hartree–Fock exchange provided the worst results, with an MUE of 0.68 *eV*. When the Hartree–Fock exchange was introduced into the functional, the MUE decreased from 0.68 *eV* to 0.39 *eV*. The MUE of PBE0, B3LYP, CAM-B3LYP, and ωB97XD were 0.57 *eV*, 0.53 *eV*, 0.49 *eV* and 0.39 *eV*, respectively. It is shown that the introduction of the Hartree–Fock exchange was helpful to reduce the effect of TDDFT on the underestimation of phosphorescence energy, and also in bringing the calculated phosphorescence energy to be closer to CC2. The MUE of the high-precision M06-2X and M06-HF was 0.29 *eV* and 0.31 *eV*, respectively. In addition, the M06-2X was functional with the closest accuracy to CC2. Therefore, the functional of the M06 series was more suitable for calculating phosphorescence energy.

The data in Table 3 show that the phosphorescence energy values were calculated by a U//UDFT fluctuate around the CC2 values. The closest to CC2 was M06-2X with an MUE of 0.14 *eV*, followed by ωB97XD with an 0.18 *eV*. With the increase in the Hartree–Fock exchange in the different functionals, the MUE values decreased from 0.66 *eV* to 0.18 *eV*. We find that increasing the percentage of the Hartree–Fock exchange significantly increased the accuracy of the functional. At this time, the MUE of ωB97XD was 0.18 *eV*, and ωB97XD was similar to that of the M06 series. The lowest MUE value of M06-2X was 0.14 *eV*, while the MUE value of M06-HF was 0.24 *eV*. The reason for this phenomenon was that M06-HF cannot calculate the three systems of BDBT, FBDBT, and BrBDBT well. Appendix A shows the result of removing the BDBT, FBDBT, and BrBDBT. At this time, the MUE of M06-HF was reduced from 0.24 *eV* to 0.06 *eV*. The MUE of M06-HF became the minimum. When M06-HF was selected as the functional, the accuracy could be improved by adding the diffuse function in a basis set, such as aug-cc-pVDZ. In Figure 1, we can see that M06-HF could achieve an ideal accuracy under the U//UDFT scheme after adding the diffuse function in the basis sets. In Table 3, we also compared the computed results to the experimental values. Generally speaking, the MUEs were relative to CC2 and the experimental values show similar trends. By using experimental values as a reference, the MUEs of PBE, PBE0, B3LYP, and CAM-B3LYP were 0.56 *eV*, 0.25 *eV*, 0.23 *eV*, and 0.22 *eV*, respectively—which were smaller than the ones that were achieved by using the CC2 as a reference. The MUEs of M06-2X and M06-HF were 0.24 *eV* and 0.46 *eV*, respectively, which are relative to the experimental values. The larger values of the MUE of M06-HF might come from the BDBT, FBDBT, and BrBDBT, as is shown in Appendix A.

**Table 3 molecules-28-03257-t003:** The energy differences ΔE (in *eV*) relative to the CC2 for the phosphorescence energy of AQs, as calculated by U//UDFT with the different functionals (PBE, PBE0, B3LYP, CAM−B3LYP, ωB97XD, M06-2X, and M06-HF) at the cc-pVTZ level.

	PBE	PBE0	B3LYP	CAM-B3LYP	ωB97XD	M06-2X	M06-HF
HA	1.47	−0.49	−0.47	−0.36	−0.31	−0.23	−0.09
AAT	−0.40	−0.41	−0.38	−0.34	−0.3	− 0.24	−0.17
BP	−0.24	−0.28	−0.18	−0.15	−0.11	−0.05	−0.02
TX−BT	−0.35	−0.34	−0.32	−0.30	−0.23	−0.07	0.14
TX-DBT	−0.62	−0.42	−0.44	−0.21	−0.14	−0.12	0.06
BDBT	−0.37	−0.28	−0.22	−0.14	−0.10	−0.03	0.75
FBDBT	−0.39	−0.29	−0.23	−0.15	−0.11	−0.04	0.75
ClBDBT	−0.39	−0.29	−0.23	−0.15	−0.10	−0.03	0.00
BrBDBT	−0.39	−0.29	−0.23	−0.15	−0.10	−0.03	0.78
XA	−0.59	−0.65	−0.53	−0.28	−0.23	−0.47	0.01
AR	1.19	−0.32	−0.44	−0.26	−0.20	−0.16	0.04
AD	1.50	−0.45	−0.45	−0.27	−0.20	−0.17	−0.02
MUE ^(a)^	0.66	0.37	0.34	0.23	0.18	0.14	0.24
MUE ^(b)^	0.56	0.25	0.23	0.22	0.21	0.25	0.46

^(a)^ MUE is the mean unsigned error between the calculated phosphorescence energy and the CC2. ^(b)^ MUE is the mean unsigned error between the calculated phosphorescence energy and the experimental value. The experimental values are shown in references [56,57,58,59,60,61,62,63].

### 3.3. Basis Sets Effect

Figure 1 shows the results obtained by four different basis sets and shows that the MUE values obtained by the PBE functional without the Hartree–Fock exchange was 0.7 *eV*. In addition, it was noted that the MUE becomes gradually smaller as the Hartree–Fock exchange increases from 0.6 *eV* to 0.18 *eV*. The results obtained using range-separated functionals were close to the best-performing M06 series. Furthermore, the difference between the ωB97XD and M06-2X was within 0.05 *eV*. The results from cc-pVDZ and cc-pVTZ were consistent. The MUE ranged from 0.7 *eV* to 0.15 *eV*, but the MUE of M06-HF rose to 0.25 *eV*. When the diffuse function was added to the basis set, the MUE of M06-HF decreased significantly from 0.25 *eV* to 0.05 *eV*, which thus became the most accurate functional. M06-HF was sensitive to the basis set, and the inclusion of the diffuse function in the basis set had a significant impact on the accuracy of the phosphorescence energy. As shown in Figure 2, we calculated the functional of M06-HF and found that the main error came from the BDBT, FBDBT, and BrBDBT in the basis set without the diffuse function.

In the M06-HF calculation with the cc-pVDZ and cc-pVTZ basis set, the errors mainly came from the BDBT, FBDBT, and BrBDBT. Appendix A shows that for the BDBT, FBDBT, and BrBDBT molecules, when using a basis set without the diffuse function, the phosphorescence energy difference between M06-HF and CC2 was over 0.8 *eV*. However, by adding the diffuse function to the basis set, it was reduced to 0.02 *eV*. As shown in Table 4 and Figure 3, we analyzed the orbitals and charge transfer. It was found that the orbitals and charge transfer that were obtained using the basis set with a diffuse function were the same as those of ClBDBT, while the orbitals of the other three systems were different, thereby leading to their large errors.

The NBO analysis was helpful to find out the types and composition of various molecular orbitals, as well as for the intramolecular and intermolecular hyperconjugation interactions. The electron withdrawing group, electron donor group, and cyclobenzene ring were the groups that often needed to be introduced in the synthesis of anthraquinone compounds. When different substituent groups were introduced, the excited states of different systems had different sensitivities when the structure changed. The order of electronic state energy levels of the anthraquinone compounds changed when compared with anthraquinone. The influence of these effects led to the rearrangement of the order of electronic state energy levels and thus affected the selection of the excited atoms for anthraquinone compounds.

Table 4 illustrates the charge transfer characteristics and the stabilization energies, E (2), of the AQs. The twelve compounds studied in this paper mainly exhibit the σ* →σ* transition form. For all the molecules except for the BDBTs (BDBT, ClBDBT, FBDBT, BrBDBT), the charge transfer characteristics and the stabilization energies were computed with the basis set of cc-pVTZ; furthermore, they were the same with the one obtained with the aug-cc-pVTZ basis set. In BDBTs, only ClBDBT has the same charge transfer characteristics and stabilization energy for both the cc-pVTZ and aug-cc-pVTZ basis sets. For the other BDBTs, i.e., in the basis set of cc-pVTZ, the charge transfer characteristics of the BDBT, FBDBT, and BrBDBT were σ* (C14-C15) → σ* (C11-C16), σ* (C14-C15) → σ * (C11-C16), and σ* (C14-C15) → σ* (C11-C16)—with stabilization energies of 516.26 *kcal/mol*, 307.23 *kcal/mol*, and 457.47 *kcal/mol*—respectively. In the aug-cc-pVTZ, charge transfer characteristics of the BDBT, FBDBT, and BrBDBT the values were σ* (C3-C4) → σ* (C1-C2), σ* (C11-C16) → σ* (C27-H31) and σ* (C22-C23) → σ* (C25-C29)—with stabilization energies of 236.54 *kcal/mol*, 1100.56 *kcal/mol*, and 441.75 *kcal/mol*—respectively. Thus, in the BDBT’s charge transfer characteristics and the stabilization energies were more sensitive to the basis set.

Here, we refer to the two singly occupied molecular orbitals (SOMOs) as SOMO1 and SOMO2. We could see that these SOMO orbitals exhibit strong charge separation properties when the diffuse function was added to the basis set. Additionally, when the basis set without the diffuse function was used, for the BDBT, FBDBT, BrBDBT, the charge separation was not as strong as it was for ClBDBT, and they were all on the same side. This is attributed to the fact that the conjugation of chlorine was better than the others, as bromine was too large, fluorine was too electronegative, and the BDBT had no substituent. This meant that the basis set with a diffuse function was very important, especially when using M06-HF. However, without the diffuse function in the basis set, the charge separation became smaller and the error increased.

## 4. Conclusions

In this work, phosphorescence energy was studied via TDDFT and UDFT with seven functionals (PEB, PBE0, B3LYP, CAM-B3LYP, ωB97XD, M06-2X, and M06-HF) and four basis sets (cc-pVDZ, cc-pVTZ, aug-cc-pVDZ, and aug-cc-pVTZ). TDDFT and UDFT were used to optimize the geometry and to calculate the phosphorescence energy, respectively. For the geometry optimization, the RMSD between the geometries optimized by TDDFT and UDFT was about 0.03 *Å*, thereby indicating that the two geometries were very close. However, as the TDDFT suffered from a Rydberg state, which was not easy to converge, UDFT was more suitable for the geometry optimization. In the phosphorescence energy calculation, when using CC2 as a reference, the MUE of UDFT was 0.14 *eV*, which is smaller than the MUE of TDDFT which was at 0.29 *eV*. Therefore, UDFT is more suitable for the calculation of phosphorescence energy. For the functional effect, we found that with increasing the Hartree–Fock exchange, the MUE decreased from 0.66 *eV* to 0.18 *eV*. Interestingly, the M06-2X functional had the smallest MUE values of 0.14 *eV*, thus indicating that the M06 series of the functionals were more suitable for describing the phosphorescence energy. More importantly, the M06-HF functional was very sensitive to the basis set. In the BDBT, FBDBT, and BrBDBT molecules, by using the basis set without the diffuse function, the phosphorescence energy difference between the M06-HF and CC2 was above 0.80 *eV*. However, by adding the diffuse function to the basis set, this decreased to 0.02 *eV*. The results of the NBO analysis on the charge transfer characteristics and stabilization energies show that the BDBT, FBDBT, and BrBDBT were more sensitive to the diffuse function of the basis set. This work has systemically studied geometry optimization and phosphorescence energy with different functionals and basis sets by using the TDDFT and UDFT methods; these findings are of particular importance for the further study of the phosphorescence energy of other organics.

## Data Availability

Not applicable.

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
