# Peer review of "Benchmark Study on Phosphorescence Energies of Anthraquinone Compounds: Comparison between TDDFT and UDFT"

_molecules, 2023, doi:10.3390/molecules28073257_

Round 1

Reviewer 1 Report

The article is an interesting study, and publishable after considering following points

1)    Authors should rewrite the abstract as the sentence are consists of four lines which gives the feeling of paragraphs

2)    Under the section geometry optimization authors mentioned some disadvantages of TDDFT in optimization of a structures. Proper citation required.

3)    For “UDFT did not have the above problems and was a more suitable method for optimization than TDDFT” authors should provide suitable references

4)    In table 1, authors compare the difference of geometry optimization between TDDFT and UDFT by calculating RMSD, but they didn’t mention the values in table are calculated at which method? Authors should have mentioned the both values calculated through TDFT and UDFT in table 1.

5)    Spin contamination existed in the UDFT, the effect of spin contamination on the phosphorescence energy was not considered here. why? Need a proper justification with suitable references

6)    Why authors used these anthraquinone compounds for current study?

7)    Are these compounds are already synthesized or they designed them

8)    To understand the charge, transfer in molecule at different basis set, authors should perform natural bond orbitals NBOs) analysis

9)    As the authors check the geometrical optimization and Phosphorescence Energy at TDDFT and UDFT, the authors should check the global reactive nature of compound. For this please check these paper (https://doi.org/10.1007/s11082-022-04441-w )

10) All the units should be italic

11) The arrangement of the manuscript is confusing. Please arrange the manuscripts according to following order: abstract, introduction, computational procedure, result and discussion and conclusion

12) The IPUAC names and structures should be mentioned at the last paragraph of introduction portion or at the first of results and discussion.

13) In computational procedure just the methodology for current study should be discussed

14) Proper citation is need

15) Please cite following reference at suitable place in manuscript

(https://doi.org/10.1016/j.molstruc.2019.127354  ,https://doi.org/10.3390/molecules24112096 )

16) The authors mentions supplementary data file, but could not find it

17) Scientific writing rules are not applied as many sentences are too lengthy just like a paragraph

Author Response

请参阅附件。

Reviewer 2 Report

The results are interesting and deserve publication, but the paper is poorly written. 

In order to calculate RMSD, the molecules need to be lined up with a proper orientation. The authors need to say how they do that.

The paper needs more careful proofreading. For example, the abstract says CC2 is a complete active space method but it is not. CC2 is an  approximate coupled-cluster singles-and-doubles model. Another example is on page 2 where it says, “Dependent density functional theory” where the meaning is “time-dependent density functional theory”. There are MANY other places where the paper has awkward phrases indicating that the authors are not fluent in English. They also confuse  singular and plural. They also do not use proper subscripts in group identifiers.

The authors should not call Rydberg states intruder states in a TDDFT calculation.  Intruder states refer to perturbation theory, not to TDDFT.

Table 1 is unclear because the authors have not yet defined the abbreviations like AAT.

The authors say that getting similar MUEs for excitation energies by two methods shows that the two methods give similar geometries. This is wrong.

Footnote b of Table 23 refers to experimental values but it does not give the experimental reference.

Author Response

Point 1: In order to calculate RMSD, the molecules need to be lined up with a proper orientation. The authors need to say how they do that.

Response 1: Thanks for the referee’s careful observation and consideration. In this work, we aligned two geometries by VMD package. For clearly, in the revised manuscript, the following sentence was added.

Line 127: “Here we aligned two geometries by visual molecular dynamics (VMD) [74] which was equivalent to minimizing the RMSD between two geometries by translating and rotating one geometry.”

Point 2: The paper needs more careful proofreading. For example, the abstract says CC2 is a complete active space method but it is not. CC2 is an approximate coupled-cluster singles-and-doubles model. Another example is on page 2 where it says, “Dependent density functional theory” where the meaning is “time-dependent density functional theory”. There are MANY other places where the paper has awkward phrases indicating that the authors are not fluent in English. They also confuse singular and plural. They also do not use proper subscripts in group identifiers.

Response 2: Thanks for the referee’s careful observation and suggestions. The use of language and phrases has been revised to make the language more rigorous. The singular and plural numbers in the text have been carefully modified. The upper and lower scripts have also been checked and modified. Meanwhile, the incorrect expression of CC2 has been modified to coupled-cluster singles and doubles, and the“Dependent density functional theory” description on page 2 has also been modified by“Density functional theory”.

Point 3: The authors should not call Rydberg states intruder states in a TDDFT calculation.  Intruder states refer to perturbation theory, not to TDDFT.

Response 3: Thanks for the referee’s suggestion. In the revised manuscript, the description of the Rydberg state is used, and the intruder state is no longer used.

Point 4: Table 1 is unclear because the authors have not yet defined the abbreviations like AAT.

Response 4: Thanks for the referee’s careful observation. The arrangement order of the manuscripts is changed, and the abbreviations are defined before discussion.

Point 5: The authors say that getting similar MUEs for excitation energies by two methods shows that the two methods give similar geometries. This is wrong.

Response 5: Thanks for the referee’s constructive suggestions. This error has been corrected in the revised manuscript.

Line 140:As shown in Table 2 and Table S18, the difference in the mean unsigned error (MUE) for the phosphorescence energy was about 0.01 eV, further proving that the geometries obtained by TDDFT and UDFT had little influence on the calculation of phosphorescence energy.

Point 6: Footnote b of Table 23 refers to experimental values but it does not give the experimental reference.

Response 6: Thanks for the referee’s careful observation. We added the reference of experimental values of AQs in footnote (b) of Table 3.

Line 211: “The experimental values are shown in references [56-63]”

Reviewer 3 Report

The work is devoted to a systematic study of the ability of time-dependent density-functional theory (TD-DFT), with a variety of exchange-correlation functionals/kernels and basis set levels, and unrestricted DFT (UDFT) to predict phosphorescence  in a series of anthraquinone compounds. Phosphorescence emission in these molecules is caused by the recombination of an electron-hole pair involving the lowest triplet excited state and the singlet ground state. The authors compare the performance of the two class of methods using high-level quantum chemistry results (complete active space with second order perturbation theory). The results indicate that, in general, UDFT is more suitable than TD-DFT for computing phosphorescence energies. A strong dependence is observed, in the case of TD-DFT, on the amount of exact (Hartree-Fock) exchange fraction in the exchange-correlation functional. Furthermore, the M06 series of functionals are shown to perform better than the other functionals considered, but also to exhibit a more marked dependence on the quality of the basis set.

The work is well carried out and clearly presented. The goals of the study are clear, and the results are well supported by the calculations. The paper reports important benchmark results which may also be helpful for future studies of classes of phosphorescence emitting molecules other than anthraquinones. I think the paper is well suited for publication in the Computational and Theoretical section of Molecules. 

Round 2

Reviewer 1 Report

Acceptable for publication now

Author Response

We kindly appreciate the referee’s recommendation for publication. 

Reviewer 2 Report

The English still needs attention. I suggest the authors obtain the help of a scientist fluent in English. Examples: “easy of fabrication” should be “ease of fabrication”. And “method of CC2” should be “method CC2”. And “single occupied” should be “singly occupied”. These are just examples. There are many places where the authors insert unnecessary “the”. There are many awkward sentences. There are also many inelegant, convoluted, and nonidiomatic sentences.

The authors say that coupled cluster theory can provide a balanced description of static and dynamic correlation.  That claim should be deleted.

DFT is claimed to provide a “perfect” balance. It is not perfect. That word should be deleted.

The subscripts are not subscripted in symmetry groups. The subscripts should be treated as subscripts.

M06-2X and M06-HF need hyphens.

The sentences “When the functional chose M06-HF, using CC2 as a reference, different basis sets were used to calculate the AQs and the error came mainly from BDBT, FBDBT and BrBDBT in the basis set without diffuse function “ and “NBO analysis was performed to investigate the interaction of charge transfer by atoms.” make no sense to me.

“seven different functionals” should be “seven functionals”.

Author Response

Point 1: The English still needs attention. I suggest the authors obtain the help of a scientist fluent in English. Examples: “easy of fabrication” should be “ease of fabrication”. And “method of CC2” should be “method CC2”. And “single occupied” should be “singly occupied”. These are just examples. There are many places where the authors insert unnecessary “the”. There are many awkward sentences. There are also many inelegant, convoluted, and nonidiomatic sentences.

Response 1: Thanks for the Reviewer’s suggestion, and we have changed them all to meet Reviewer’s thoughts. We changed “method of CC2” to “method CC2”, “single occupied” to “singly occupied” and removed the extra "the". We modified the language according to the rules of scientific writing to make the language description more scientific. We sought help with professional English language modifications, and the certificate was uploaded in the attachment. We revised the use of terminology and language as requested by the reviewers.

Point 2: The authors say that coupled cluster theory can provide a balanced description of static and dynamic correlation.  That claim should be deleted.

Response 2: Thanks for the Reviewer’s comments. In the revised manuscript, we rewrote this sentence to delete the expression of static correlation.

Line 47: “Thus, the coupled‐cluster method, as a high level ab initio method including dynamic correlation, are usually adopted for the calculation of phosphorescence energy.”

Point 3: DFT is claimed to provide a “perfect” balance. It is not perfect. That word should be deleted.

Response 3: Thanks for the Reviewer’s comment. In the revised manuscript, we deleted the expression of "perfect".

Point 4: The subscripts are not subscripted in symmetry groups. The subscripts should be treated as subscripts.

Response 4: We modified the representation of symmetry by changing it to subscript and changing “Cs” to “Cs”, “C1” to “C1

Point 5: M06-2X and M06-HF need hyphens.

Response 5: Thanks for the Reviewer’s suggestion, we have modified the names of these functionals in our manuscript.

Point 6: The sentences “When the functional chose M06-HF, using CC2 as a reference, different basis sets were used to calculate the AQs and the error came mainly from BDBT, FBDBT and BrBDBT in the basis set without diffuse function “and “NBO analysis was performed to investigate the interaction of charge transfer by atoms.” make no sense to me.

Response 6: Thanks for the Reviewer’s suggestion. We have modified these sentences and deleted meaningless sentences.

Line 234: We rewrote this “In M06-HF calculation with cc-pVDZ and cc-pVTZ basis set, the errors mainly came from BDBT, FBDBT and BrBDBT.

Line 246: We delete this “NBO analysis was performed to investigate the interaction of charge transfer by atoms.

Point 7: “seven different functionals” should be “seven functionals”.

Response 7: Thanks for the Referee’s careful observation and suggestions. We changed “seven different functionals” to “seven functionals”
